# Sublimation Study of Six 5-Substituted-1,10-Phenanthrolines by Knudsen Effusion Mass Loss and Solution Calorimetry

**DOI:** 10.3390/e24020192

**Published:** 2022-01-27

**Authors:** Bruno Brunetti, Andrea Ciccioli, Andrea Lapi, Aleksey V. Buzyurov, Ruslan N. Nagrimanov, Mikhail A. Varfolomeev, Stefano Vecchio Ciprioti

**Affiliations:** 1Istituto per lo Studio dei Materiali Nanostrutturati, Consiglio Nazionale delle Ricerche, Dipartimento di Chimica, Sapienza Università di Roma, P.le A. Moro 5, 00185 Rome, Italy; bruno.brunetti@cnr.it; 2Dipartimento di Chimica, Sapienza Università di Roma, P.le A. Moro 5, 00185 Rome, Italy; andrea.ciccioli@uniroma1.it (A.C.); andrea.lapi@uniroma1.it (A.L.); 3Istituto per i Sistemi Biologici (ISB-CNR), Sede Secondaria di Roma-Meccanismi di Reazione, Dipartimento di Chimica, Università degli Studi di Roma “La Sapienza”, P.le A. Moro 5, 00185 Rome, Italy; 4A.M. Butlerov Chemical Institute, Kazan Federal University, Kremlevskaja Str. 18, 420008 Kazan, Russia; abuzurov95@gmail.com (A.V.B.); rnagrimanov@gmail.com (R.N.N.); 5Department of Petroleum Engineering, Institute of Geology and Petroleum Technologies, Kazan Federal University, Kremlevskaja Str. 18, 420008 Kazan, Russia; 6Dipartimento di Scienze di Base ed Applicate per l’Ingegneria (S.B.A.I.), Sapienza University of Rome, Via del Castro Laurenziano 7, Building RM017, 00161 Rome, Italy

**Keywords:** 5-substituted-1,10-phenanthrolines, vapor pressure, Knudsen Effusion, solution calorimetry, standard molar sublimation enthalpy, standard molar sublimation entropy, standard molar sublimation Gibbs energies, entropic effect of rotation

## Abstract

The vapor pressures of six solid 5-X-1,10-phenanthrolines (where X = Cl, CH_3_, CN, OCH_3_, NH_2_, NO_2_) were determined in suitable temperature ranges by Knudsen Effusion Mass Loss (KEML). From the temperature dependencies of vapor pressure, the molar sublimation enthalpies, Δ_cr_^g^*H*_m_^0^(⟨*T*⟩), were calculated at the corresponding average ⟨*T*⟩ of the explored temperature ranges. Since to the best of our knowledge no thermochemical data seem to be available in the literature regarding these compounds, the Δ_cr_^g^*H*_m_^0^(⟨*T*⟩) values obtained by KEML experiments were adjusted to 298.15 K using a well known empirical procedure reported in the literature. The standard (*p*^0^ = 0.1 MPa) molar sublimation enthalpies, Δ_cr_^g^*H*_m_^0^(298.15 K), were compared with those determined using a recently proposed solution calorimetry approach, which was validated using a remarkable amount of thermochemical data of molecular compounds. For this purpose, solution enthalpies at infinite dilution of the studied 5-chloro and 5-methylphenantrolines in benzene were measured at 298.15 K. Good agreement was found between the values derived by the two different approaches, and final mean values of Δ_cr_^g^*H*_m_^0^(298.15 K) were recommended. Finally, the standard molar entropies and Gibbs energies of sublimation were also derived at *T* = 298.15 K. The volatilities of the six compounds were found to vary over a range of three orders of magnitude in the explored temperature range. The large difference in volatility was analyzed in the light of enthalpies and entropies of sublimation. The latter was tentatively put in relation to the rotational contribution of the substituent group on the phenanthroline unit.

## 1. Introduction

The aromatic heterocyclic compound 1,10-Phenanthroline is formed by three condensed rings (Figure 1), and is well known for its pronounced metal-ion complexing properties [1,2,3,4,5,6].

Several years ago, the molar heat capacities and some thermodynamic properties of phen, its 2,9-dimethyl derivative and polycyclic related compounds were determined in a suitable range of temperatures [7,8,9,10,11]. More recently, [12] we determined for the first time the enthalpies and entropies of sublimation/evaporation of 1,10-phenanthroline from vapor pressure measurements performed by the Knudsen effusion mass loss (KEML) technique on the solid phase and isothermal thermogravimetry (TG) on the liquid phase, supported by differential scanning calorimetry (DSC) to determine the enthalpy of fusion. Before that work, the only information available on thermodynamic properties of different phen derivatives, denoted as three-ring aza-aromatics, were on their sublimation enthalpies [13,14].

As a continuation of our work on tricyclic nitrogen heterocyclic compounds, we have undertaken a systematic study of the sublimation thermodynamics of 1,10-phenanthroline derivatives, for which thermodynamic information is extremely scarce. In particular, in this paper we focused on six 5-substituted 1,10-phenanthrolines, for which no thermodynamic information is presently available. The molecular structure of the compounds tested is reported in Figure 1. KEML measurements were performed on the crystal phases of these compounds and sublimation enthalpies derived. Furthermore, in the effort to perform a mutual validation of two completely independent techniques, the solution calorimetry method developed by Solomonov and co-workers [15,16] was also used for two compounds. This method is based on the relationship between evaporation/sublimation enthalpy and enthalpies of solution and solvation of a studied compound in a properly selected solvent. The main advantage of this method is that it is able to provide the stan-dard molar evaporation or sublimation enthalpy (above a liquid or solid, respectively) at 298.15 K by measuring the solution enthalpy of the compound tested in a selected solvent and using a group-additivity scheme for calculation of the solvation enthalpy, without any enthalpic contribution due to adjustment to the reference temperature.

It is well known that, while measuring vapor pressures is the most direct procedure to derive sublimation enthalpies, a reliable determination of this thermodynamic parameter at a reference temperature *T*_ref_ can be completed. To this end, some experiments were performed using two or more different techniques in suitable temperature ranges above the same phase or on two different phases (i.e., above the solid and the liquid), as undertaken by our group in the recent past [17,18], as well as by using some additivity properties referred to as *T*_ref_ (without the need to consider any contribution due to its adjustment to *T*_ref_), especially when no literature data are available (as in this case, according to our knowledge).

So, the aim of this study is to report the first vapor pressure and thermodynamic data on sublimation for six solid 5-x-1,10-phenanthrolines (where X = Cl, CH_3_, CN, OCH_3_, NH_2_, NO_2_) and to cross-validate with calorimetric results, with the view to guarantee the accuracy of the derived thermochemical properties.

## 2. Materials and Methods

### 2.1. Compounds

Source, purification method and mass fraction purity of the studied compounds are reported in Table 1.

In particular, 5-Cl- and 5-CH_3_-1,10-phenanthrolines were purchased (Sigma-Aldrich, purity > 98%) and used as received. The 1,10-Phenanthroline-5,6-epoxide, precursor for the synthesis of 5-CH_3_O-1,10-phenanthroline and 5-CN-1,10-phenanthroline, was prepared from reaction of 1,10-phenanthroline and sodium hypochlorite according to a literature procedure [19]. The synthesis route for 5-CN-1,10-phenanthroline, 5-CH_3_O-1,10-phenanthroline, 5-NO_2_-1,10-phenanthroline and 5-NH_2_-1,10-phenanthroline is reported in Figure 2, while all details for their preparation are shown in the Appendix A.

The synthesis of the 5-substituted nitro and amino derivatives (Figure 2c,d) was made according to a procedure reported in reference [20]. Water content in 5-Cl- and 5-CH_3_-1,10-phenanthrolines was checked using Karl Fischer titration (Mettler Toledo C20 Coulometric KF Titrator).

### 2.2. Instruments

#### 2.2.1. DSC Measurements

The melting temperatures and the enthalpies of fusion have been determined using a STA625 Stanton Redcroft simultaneous TG/DSC apparatus, consisting of two open cylindrical shape aluminum pans, one empty for the reference and the other filled with a suitable amount of sample. DSC experiments were carried out under argon flow rate of 20 mL·min^−1^ at 2 K·min^−1^ and the raw data were acquired using a personal computer through the RSI Orchestrator software supplied by Rheometric Scientific. Calibration of temperature and heat flux was made using recommended high purity reference materials (benzoic acid and indium in this study), whose melting temperatures *T*_fus_ and enthalpies of fusion Δ_cr_^l^*H*_m_^0^ (*T*_fus_) are well known [21,22]. Based on these calibrations, we estimated u(*T*_fus_) = 2·u(*T*) = 0.2 K, while for the Δ_cr_^l^*H*_m_^0^(*T*_fus_) the standard deviations of three replicates combined with the uncertainty of the heat flow calibration have been considered more appropriate.

#### 2.2.2. KEML Measurements

The Knudsen effusion mass loss (KEML) experiments were carried out using a Ugine-Eyraud Model B60 Setaram thermobalance, accurately described in a previous paper [23]. It is essentially constituted by a furnace, a microbalance and a vacuum system. The measuring cell is housed in a copper cylinder with a cap, which has the purpose of equalizing the temperature of the sample to allow an optimal temperature measurement. The copper cylinder is suspended to the arm of the microbalance with a standard measurement uncertainty u(*m*) = 0.01 mg. The temperature was measured via a Pt100 Platinum Resistance Thermometer inserted into the copper cylinder, being the standard measurements uncertainty u(*T*) less than 0.2 K. The temperature control and the measurements of the mass loss are made through a data logger (HP 34970A) driven by a LabVIEW software that permits the continuous control of the system.

Two alumina cells (A and B) with different effusion orifice diameters (OD) of 3 and 1 mm, respectively, were alternatively used by loading approximately 50 mg of sample, previously purified by sublimation under reduced pressure. The instrumental Knudsen constant [24] was evaluated for each cell by performing KEML experiments (series A with OD = 3 mm and series B with OD = 1 mm) under identical conditions of the compounds tested using very pure calibration substances having well known vapor pressures (benzoic acid [25,26] in this study). During each KEML experiment the temperature of the sample was adjusted to evaluate the mass loss rate at different constant temperatures. For each experiment, the isothermal temperature explored was first decreased and then increased. This approach allows detecting possible changes in the composition of the sample caused by impurities or decomposition reactions. This procedure would produce a gradual variation of the sample’s vapor pressures, and, therefore, two non-overlapping data sets.

#### 2.2.3. Solution Calorimetry Measurements

The solution enthalpies of 5-chloro and 5-methyl-1,10-phenanthrolines in benzene at infinite dilution were measured at *T* = 298 K in a concentration range from 1.6 to 7.13 mmol·kg^−1^ using a TAM III precision solution calorimeter (Appendix A).

The samples were dissolved by breaking a glass ampule in a glass cell containing pure benzene. The details of the solution calorimetry experimental procedure have been fully described elsewhere [27].

## 3. Results and Discussion

### 3.1. Melting Parameters Determination by DSC Experiments

The melting temperatures and the enthalpies of fusion of the six 5-X-1,10-phenanthroline derivatives are reported in Table 2, along with the associated uncertainties. At the present time, according to our knowledge, no melting data concerning these compounds are available in literature.

### 3.2. Vapor Pressure Determination by KEML Experiments

The experimental vapor pressure data determined by KEML above the solids in suitable experimental temperature ranges are summarized in Table 3. The corresponding plots are shown in Figure 3.

### 3.3. Sublimation Enthalpies of 1,10-Phenanthrolines Obtained by the Solution Calorimetry Approach

Determination of sublimation enthalpies of 5-chloro and 5-methyl-1,10-phenanthrolines is a challenging task since no values are now available in the literature, and it has prompted us to apply a solution calorimetry method for these compounds. The approach for determination of evaporation and sublimation enthalpies based on the relationship between evaporation/sublimation enthalpy of a compound *A*_i_, with its solution Δ_soln_*H*_m_^*A*_i_^^/*S*^ and solvation Δ_solv_*H*_m_^*A*_i_^^/S^ enthalpies in a solvent S was developed and validated in [15,16,27,28]:Δ_cr_^g^*H*_m_^0^ = Δ_soln_*H*_m_^*A*_i_^^/*S*^ − Δ_solv_*H*_m_^*A*_i_^^/*S*^,(1)

Enthalpies of solution of 5-chloro and 5-methyl-1,10-phenanthroline in benzene were measured by using solution calorimetry (see Appendix A).

Enthalpies of solvation were predicted by using an additive scheme described elsewhere [16,28]. According to this approach, solvation enthalpy of substituted 1,10-phenanthrolines in benzene can be calculated using the following equation:Δ_solv_*H*_m_^*A*_i_^^/*S*^ = Δ_solv_*H*_m_*^ArH^*^/*S*^ + Σ Δ_solv_*H*_m_^*Y*_i_^^→*CH*/*S*^ + Σ Δ_solv_*H*_m_^*X*_i_^^→*H*/*S*^,(2)
where Δ_solv_*H*_m_*^ArH^*^/*S*^ is the solvation enthalpy of a parent aromatic compound, Δ_solv_*H*_m_^*X*_i_^^→*H*/*S*^ is the contribution to the solvation enthalpy related to a replacement of the hydrogen atoms by any other groups, and Δ_solv_*H*_m_^*Y*_i_^^→*CH*/*S*^ is a contribution to the solvation enthalpy related to a replacement of the *CH*-group in aromatic rings by any other groups. In the case of the substituted 1,10-phenanthrolines the solvation enthalpies in benzene were calculated as a sum of solvation enthalpy of phenanthrene (−74.6 kJ·mol^−1^) [28] and the contribution related to the replacement of the CH-fragment in these rings by the specific units for 1,10-phenanthrolines: −N = (−5.4 kJ·mol^−1^) [16] and the contribution related to the replacement of the H-atom by the CH_3_—group (−3.5 kJ·mol^−1^), and Cl—group (−6.1 kJ·mol^−1^) [15].

Results of calculations of the Δ_cr_^g^*H*_m_^0^ values according to Equations (1) and (2) based on experimental solution enthalpies are given in Table 4.

### 3.4. Standard Molar Sublimation Enthalpies

The *p*/Pa vs. K/*T* data obtained from KEML experiments on effusion holes of 1- and 3-mm diameters have been reported in Figure 3, and the fitted regression parameters obtained from the least square method, intercepts and slopes (A and B, respectively), along with the associated uncertainties as standard deviations, are given in Table 5.

From the slopes *B* of these regression lines, the corresponding molar sublimation enthalpies, at the mean values ⟨*T*⟩ of the corresponding experimental temperature ranges, Δ_cr_^g^*H*_m_^0^(⟨*T*⟩) have been obtained as the absolute value of the product of the slope and the gas constant *R*. The values are reported in Table 6.

In order to adjust the sublimation enthalpy values to the common reference temperature of 298 K, the following empirical formula proposed by Chickos et al. [29,30] was used:Δ_cr_^g^*H*_m_^0^(298.15 K)/kJ·mol^−^^1^ = Δ_cr_^g^*H*_m_^0^(⟨*T*⟩) + [Δ_cr_^g^*C*_p,m_^0^/J·K^−^^1^·mol^−^^1^]·(⟨*T*⟩/K − 298.15)/1000,(3)
where [Δ_cr_^g^*C*_p,m_^0^/J·K^−1^·mol^−1^] = −[0.75 + 0.15·*C*_p,m_^0^ (cr)], being *C*_p,m_^0^ (cr), the isobaric standard molar heat capacities of the crystals at the reference temperature (not available in the literature). The *C*_p,m_^0^ values estimated by the group additivity contribution method using the functional group values given in [31] and the Δ_cr_^g^*C*_p,m_^0^ values have been also reported in Table 6.

The standard molar sublimation enthalpies referenced to *T* = 298.15 K, Δ_cr_^g^*H*^0^_m_(298.15 K), calculated according to Equation (3) are reported in Table 6, along with those obtained using the solution calorimetry. It is worth noting that in all cases the values derived from KEML experiments with different effusion orifice diameters are in agreement within the corresponding error interval. Comparison of the sublimation enthalpies derived by the solution calorimetry approach and data measured by using KEML (Table 6) shows good agreement within the boundaries of the experimental uncertainties. Such a good agreement between different methods can be considered as evidence of the consistency of the sublimation data for 5-Cl- and 5-CH_3_-1,10-phenanthrolines in Table 6. For future thermochemical calculations, average sublimation enthalpies from two methods should be used.

The vapor pressure values at the average temperature *T* = ⟨*T*⟩, *p*(⟨*T*⟩), calculated using the regression parameters A and B displayed in Table 5, are given in Table 7, along with the molar sublimation entropies at ⟨*T*⟩ and *p*(⟨*T*⟩): Δ_cr_^g^*S*_m_^0^ (⟨*T*⟩, *p*(⟨*T*⟩). The standard molar sublimation entropies values Δ_cr_^g^*S*^0^_m_(298.15 K, *p*^0^), calculated at *T* = 298.15 K and at *p*^0^ = 0.1 MPa, are determined using the following equation
Δ_cr_^g^*S*_m_^0^(298.15 K, *p*^0^)/J· K^−1^·mol^−1^ = Δ_cr_^g^*S*_m_^0^(⟨*T*⟩, *p*⟨*T*⟩)/J·K^−1^·mol^−1^ + Δ_cr_^g^*C*_p,m_^0^/J·K^−1^·mol^−1^ · ln[(298.15 K/⟨*T*⟩)] − *R*·ln(*p*^0/^*p*⟨*T*⟩)(4)

The values of both the standard molar sublimation entropies and Gibbs energy, Δ_cr_^g^*S*^0^_m_(298.15 K, *p*^0^) and Δ_cr_^g^*G*^0^_m_(298.15), respectively, are listed in Table 7, while the uncertainties due to the adjustment to 298 K were taken, as suggested in the literature [31], as one third of the correction itself.

The analysis of Figure 3 and Table 6 shows several interesting results. First, the volatilities of the 5-substituted 1,10-phenanthrolines here under study are very different depending on the substituent. In particular, the volatility of the amino-substituted compound is much lower than the cyano- and nitro-derivatives (whose vapor pressures are very similar) and, even more, lower than the methoxy-, methyl- and chloro- ones. In the temperature range explored in our experiments, the presence of the amino group leads to a vapor pressure by 20 to 40 times lower compared with cyano- and nitro-substituted 1,10-phenanthrolines. This difference is in large part associated with the high enthalpy of sublimation (Table 6), most probably due to the hydrogen bonds established by the amino group in the crystal phase, as observed by X-ray diffraction in aniline [33].

In the explored temperature range, the vapor pressure of the methoxy-substituted derivative is about 14–16 times higher than that of the corresponding cyano- and nitro- compounds, despite a very similar sublimation enthalpy. In this case the difference in volatility is mainly due to the entropic term, which is significantly higher for 5-methoxy-1,10-phenanthroline. In the case of the chloro-and methyl-derivative, which are the most volatile compounds in this group, the lowest enthalpy of sublimation largely overcomes the effect of the low entropy change. It is interesting to wonder to what extent the differences between the sublimation entropies of the six compounds (Table 7) may be read in the light of the rotation of the group in position five, which is expected to be enhanced in the gas phase compared to the crystal [34]. Indeed, this can account for the low value of the chloro-derivative, whereas the fairly large value of the cyano-derivative calls for a different explanation. In the case of the nitro- group, the partially double nature of the C–N bond could lower the gain in the rotational entropy compared to the methoxy-substituted compound, making the latter significantly more volatile, as mentioned above.

## 4. Conclusions

The Knudsen Effusion Mass Loss (KEML) technique was used for the first time to determine the vapor pressures of the six 5-substituted 1,10-phenanthrolines (with the following substituents: Cl, CH_3_, CN, OCH_3_, NH_2_, NO_2_. The molar sublimation enthalpies at the corresponding average temperature ⟨*T*⟩, were calculated from the temperature dependencies of vapor pressure. These values were adjusted to 298.15 K using a well known empirical procedure reported in the literature, and the standard (*p*^0^ = 0.1 MPa) molar sublimation enthalpies Δ_cr_^g^*H*_m_^0^(298.15 K) were compared with those determined using solution calorimetry, according to a procedure recently proposed for validation purpose. Good agreement was found between the values derived by the two different approaches. Finally, the standard molar entropies and Gibbs energies of sublimation were also derived at *T* = 298.15 K.

The volatility of the six compounds was found to vary over a range of three orders of magnitude going from the less volatile amino derivative to the most volatile methyl and chloro-substituted compounds. The observed trend of volatility amino < cyano ≅ nitro < methoxy < methyl ≅ chloro parallels, as expected, that of the corresponding sublimation enthalpies, with the exception of the methoxy derivative, whose volatility is one order of magnitude greater than that of the cyano and nitro compounds, in spite of a very similar sublimation enthalpy. An explanation of the increased volatility of the methoxy compound can be found in its larger sublimation entropy, possibly related in turn to the increase of the rotational entropy of the substituent group from the crystal to the gas phase. Indeed, the more the substituents are free to rotate in the gas phase compared to the crystal phase, the larger will be the related entropy gain due to sublimation. This might explain the low value of the sublimation entropy of the chloro-derivative and the larger values of the methoxy- and amino- compounds. A lower rotational gain could be speculated for the nitro- group due to the partially double nature of C–N bond that hinders the rotation of the substituent group of the molecule in the gaseous phase.

## Figures and Tables

**Figure 1 entropy-24-00192-f001:**
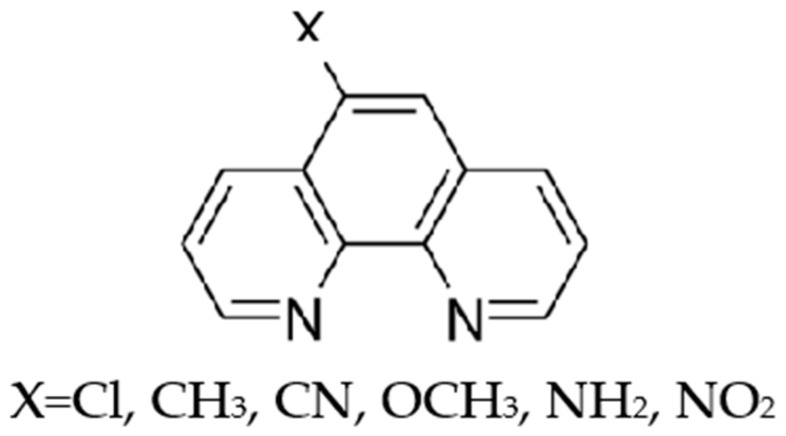
Molecular structure of the 5-substituted 1,10-phenanthrolines.

**Figure 2 entropy-24-00192-f002:**
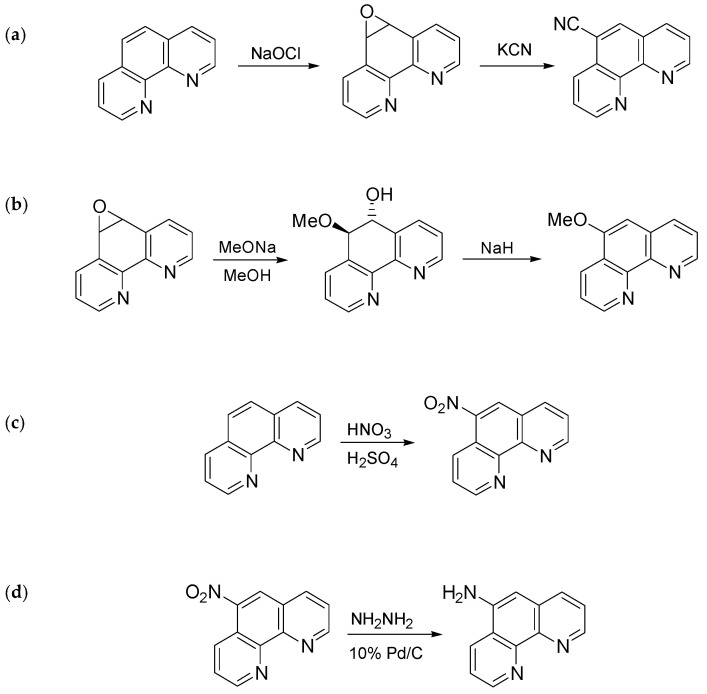
Synthesis route of: 5-cyano-1,10-phenanthroline (**a**), 5-methoxy-1,10-phenanthroline (**b**), 5-nitro-1,10-phenanthroline (**c**) and 5-amino-1,10-phenanthroline (**d**).

**Figure 3 entropy-24-00192-f003:**
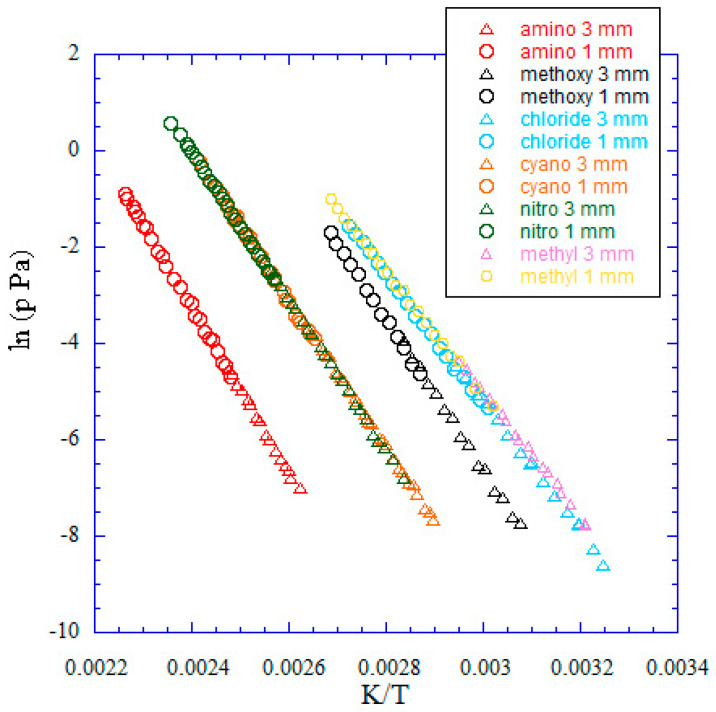
Vapor pressures of the six 5-substituted 1,10-phenanthrolines vs. temperature, measured by KEML.

**Table 1 entropy-24-00192-t001:** Source, purification method and mass fraction purity of the studied compounds.

Compound	Source	Purification Method	Final Mass Fraction Purity	Analysis Method	Water Content/ppm
5-Cl-1,10-phenanthroline	Sigma-Aldrich	-	>0.98	-	50
5-CH_3_-1,10-phenanthroline	Sigma-Aldrich	-	>0.99	-	30
5-CH_3_O-1,10-phenanthroline	Synthesis	Chromatography	>0.999	GC	-
5-CN-1,10-phenanthroline	Synthesis	Recrystallization	>0.999	GC	-
5-NO_2_-1,10-phenanthroline	Synthesis	Recrystallization	>0.999	GC	-
5-NH_2_-1,10-phenanthroline	Synthesis	Recrystallization	>0.999	GC	-
benzene	Ekos-1	Distillation	>0.999	GC	20

**Table 2 entropy-24-00192-t002:** Melting (onset) temperatures and molar enthalpies of fusion of the compounds studied determined by DSC.

Compound	*T* _fus_ ^a^	Δ_cr_^l^*H*_m_^0 b^
	K	kJ·mol^−1^
5-Cl-1,10-phenanthroline	396.5 ± 0.2	18.9 ± 0.6
5-CH_3_-1,10-phenanthroline	384.3 ± 0.2	8.9 ± 0.4
5-CH_3_O-1,10-phenanthroline		
5-CN-1,10-phenanthroline		
5-NO_2_-1,10-phenanthroline	473.0 ± 0.2	25.2 ± 0.8
5-NH_2_-1,10-phenanthroline	525.1 ± 0.2	24.1 ± 0.8

^a^ Uncertainty of melting temperature corresponds to twice the standard uncertainty of calibration temperature (u(*T*_fus_)) = 2·u(*T*)). ^b^ Uncertainty related to the enthalpy of fusion combines the standard deviation of three replicates with the uncertainties of the heat flow calibration.

**Table 3 entropy-24-00192-t003:** Vapor pressures of crystalline 5-substituted-1,10-phenanthrolines measured by KEML using two different effusion orifices (1 and 3 mm diameter).

*T*/K	Δ*t*/s	Δ*m*/mg ^a^	*p*/Pa ^a^	100 Δ*p*/*p* ^b^	*T*/K	Δ*t*/s	Δ*m*/mg ^a^	*p*/Pa ^a^	100 Δ*p*/*p* ^b^
5-chloro-1,10-phenanthroline									
*∮*/mm = 1					*∮*/mm = 3				
367.1	4074	1.20	0.206	−3.7	323.1	15,821	0.19	0.00152	−2.6
361.3	4430	0.77	0.121	2.7	312.9	40,125	0.14	0.000437	7.8
355.4	4593	0.42	0.0626	−2.0	332.7	6951	0.29	0.00535	3.7
349.6	6239	0.30	0.0328	−4.0	327.7	14,246	0.31	0.00278	−0.3
343.8	9256	0.23	0.0167	−6.0	322.6	24,318	0.29	0.00153	4.4
338.0	21,233	0.29	0.00907	0.8	317.8	27,704	0.17	0.000784	0.9
332.2	31,710	0.23	0.00479	6.2	312.9	32,694	0.11	0.000424	4.8
365.2	5447	1.39	0.178	1.0	308.0	96,302	0.14	0.000186	−10.0
359.3	4836	0.67	0.0963	0.4	339.6	7394	0.65	0.0115	−1.0
353.6	4720	0.37	0.0537	2.7	337.6	6903	0.48	0.00903	−2.5
347.7	9264	0.37	0.0274	−0.2	334.7	9694	0.47	0.00624	−5.2
341.9	12,381	0.26	0.0142	−0.3	329.8	15,100	0.46	0.00389	6.4
336.1	24,654	0.26	0.00705	−1.8	325.0	19,951	0.30	0.00192	−3.4
363.3	5313	1.15	0.151	4.1	320.1	37,451	0.31	0.00104	−2.7
357.4	4453	0.51	0.0796	0.9	315.3	52,795	0.23	0.000552	−0.9
351.6	6174	0.38	0.0419	−0.9	309.8	72,493	0.15	0.000260	−1.1
345.8	12,210	0.41	0.0227	2.5					
339.9	27,239	0.44	0.0108	−5.0					
334.1	58,330	0.50	0.00577	1.5					
5-methyl-1,10-phenanthroline									
*∮*/mm = 1					*∮*/mm = 3				
372.3	5983	2.72	0.373	2.9	337.0	9042	0.59	0.0107	2.1
368.4	4837	1.48	0.250	−0.3	334.0	8058	0.37	0.00749	0.7
364.5	4987	1.03	0.168	−2.2	331.1	10,234	0.33	0.00516	−1.8
360.5	5436	0.80	0.119	2.2	328.1	13,521	0.31	0.00365	−1.4
356.6	5022	0.49	0.0785	0.0	325.2	22,546	0.35	0.00252	−2.4
352.7	6864	0.47	0.0542	2.7	322.2	21,835	0.25	0.00179	0.0
348.8	9740	0.44	0.0361	3.7	319.3	48,156	0.39	0.00128	3.3
344.7	17,221	0.48	0.0220	−2.6	316.3	66,072	0.35	0.000837	−1.4
340.8	18,658	0.33	0.0141	−4.1	311.6	244,549	0.69	0.000440	−3.9
370.3	5857	2.15	0.301	−0.1	338.9	7466	0.59	0.0128	−1.3
366.4	4954	1.25	0.206	−0.8	335.0	5878	0.29	0.00817	−1.5
362.4	4814	0.86	0.144	2.8	332.0	7950	0.29	0.00585	−0.5
358.5	6889	0.82	0.0955	0.4	329.1	12,120	0.31	0.00417	0.6
354.6	6808	0.51	0.0604	−6.0	326.1	19,705	0.34	0.00275	−4.8
350.7	9825	0.50	0.0403	−5.8	323.1	20,423	0.28	0.00219	8.4
346.8	12,692	0.45	0.0283	0.0	320.2	42,259	0.37	0.00139	0.4
342.9	15,825	0.37	0.0186	0.8	317.2	57,261	0.36	0.00101	5.6
338.9	22,030	0.35	0.0125	5.0	314.3	87,374	0.36	0.000646	−0.8
335.0	39,950	0.37	0.00726	−4.5	311.4	139,369	0.38	0.000431	−2.3
331.0	54,412	0.35	0.00504	5.0					
5-methoxy-1,10-phenanthroline									
*∮*/mm = 1					*∮*/mm = 3				
372.3	6709	1.52	0.183	0.6	352.6	6056	0.90	0.0194	4.5
368.4	4712	0.70	0.119	2.9	348.4	9816	0.86	0.0113	5.5
364.5	4606	0.43	0.0757	3.7	344.4	4851	0.25	0.00658	3.7
360.4	6832	0.39	0.0456	2.1	340.6	6547	0.20	0.00390	2.8
356.3	9829	0.35	0.0285	5.4	336.7	10,399	0.18	0.00225	2.3
352.3	15,280	0.33	0.0168	3.4	332.8	27,497	0.29	0.00134	5.5
348.5	40,339	0.51	0.00984	0.0	328.9	48,497	0.29	0.000758	5.1
370.7	5230	0.93	0.144	−4.5	325.0	111,603	0.39	0.000434	7.2
366.7	4490	0.52	0.0930	−2.5	350.4	5903	0.61	0.0134	−3.1
362.3	6220	0.42	0.0544	−2.6	346.5	4992	0.31	0.00801	−3.7
358.3	9393	0.40	0.0335	−2.5	342.6	6606	0.24	0.00471	−4.7
354.3	15,204	0.40	0.0208	−0.6	338.7	15,350	0.32	0.00268	−7.9
350.4	24,988	0.38	0.0120	−6.1	334.5	31,848	0.37	0.00148	−9.0
					330.6	50,156	0.34	0.000865	−6.2
					326.7	48,936	0.19	0.000496	−4.0
5-cyano-1,10-phenanthroline									
*∮*/mm = 1					*∮*/mm = 3				
413.7	4686	4.33	0.799	1.5	376.4	6387	1.01	0.0217	6.4
409.8	4660	2.80	0.518	−5.7	372.4	4140	0.43	0.0141	7.6
405.9	4651	2.10	0.387	3.1	368.5	5050	0.32	0.00865	3.3
401.9	4673	1.43	0.261	2.1	364.7	8911	0.37	0.00552	2.5
398.1	4737	0.99	0.177	1.4	360.8	17,601	0.46	0.00349	2.8
394.2	4736	0.67	0.119	0.8	357.0	21,827	0.37	0.00226	6.1
390.3	4388	0.42	0.0801	1.3	353.0	39,028	0.37	0.00127	−2.6
386.3	6370	0.40	0.0527	1.0	349.2	69,135	0.43	0.000810	1.6
382.5	9638	0.38	0.0327	−5.8	345.3	11,8495	0.42	0.000463	−4.6
378.7	19,865	0.57	0.0236	3.6	378.3	6794	1.16	0.0234	−7.7
408.3	5201	2.84	0.468	−1.0	374.4	5730	0.68	0.0161	−1.4
404.3	4650	1.71	0.314	−3.1	370.6	4833	0.36	0.0101	−4.5
402.0	5241	1.49	0.242	−6.6	366.6	9197	0.46	0.00680	1.1
398.0	4350	0.84	0.164	−5.8	362.7	17,937	0.56	0.00421	−1.0
400.5	5104	1.31	0.219	−2.2	358.9	21,220	0.41	0.00257	−4.1
396.5	4545	0.79	0.147	−1.9	355.0	38,864	0.48	0.00164	−1.6
392.7	4333	0.51	0.0991	−2.5	351.1	69,000	0.52	0.00100	−2.5
388.8	4513	0.36	0.0665	−2.3	347.2	94,468	0.43	0.000598	−4.0
384.8	6556	0.34	0.0430	−3.2	377.1	4614	0.74	0.0218	−0.6
381.0	9294	0.32	0.0285	−3.0	369.4	4526	0.30	0.00909	−2.0
377.1	18,437	0.45	0.0203	5.8	373.3	6353	0.65	0.0140	−2.4
408.5	5599	3.18	0.488	1.1	365.6	8871	0.38	0.00584	−1.8
404.6	5152	2.07	0.344	3.1	361.7	18,077	0.49	0.00363	−2.9
400.8	5450	1.62	0.253	8.9	357.8	21,203	0.39	0.00247	4.8
396.9	4550	0.89	0.165	5.5	353.9	39,299	0.40	0.00135	−7.1
393.1	5393	0.70	0.110	4.1	350.0	68,371	0.50	0.000967	8.9
389.2	5304	0.46	0.0725	2.7	346.0	72,267	0.30	0.000551	3.4
385.4	8506	0.46	0.0454	−3.9					
381.4	6767	0.25	0.0306	−1.1					
5-nitro-1,10-phenanthroline									
*∮*/mm = 1					*∮*/mm = 3				
411.9	4784	4.41	0.638	1.5	392.2	3973	2.93	0.100	−2.5
408.9	3676	2.55	0.480	−0.1	387.1	4240	1.92	0.0612	0.9
405.9	4039	2.15	0.366	1.4	382.6	2480	0.71	0.0382	2.0
403.0	4165	1.68	0.276	0.4	377.7	4454	0.75	0.0225	2.7
399.9	4172	1.25	0.204	−0.4	372.3	2679	0.25	0.0123	3.0
396.9	4283	0.95	0.151	−0.7	367.3	10,640	0.56	0.00698	3.5
394.3	4363	0.74	0.115	−1.5	362.4	20,845	0.61	0.00382	1.3
391.0	4053	0.50	0.0834	−1.1	357.7	44,081	0.72	0.00212	−0.4
389.0	4737	0.48	0.0685	−0.7	352.6	141,123	1.24	0.00113	0.7
413.2	4482	4.60	0.712	0.0	390.1	4700	2.85	0.0821	−0.8
410.2	4535	3.55	0.540	−0.3	385.2	4181	1.50	0.0485	−1.9
407.3	4522	2.73	0.416	0.9	380.1	4440	0.97	0.0291	1.9
404.5	4632	2.15	0.319	1.2	375.2	3939	0.50	0.0170	2.4
401.5	4598	1.59	0.237	−0.3	370.3	6481	0.47	0.00962	1.1
398.5	4622	1.20	0.177	−0.4	365.4	12,680	0.51	0.00524	−3.2
395.6	4524	0.89	0.134	0.7	360.4	21,482	0.45	0.00274	−7.7
392.6	4533	0.68	0.101	1.4	355.4	71,928	0.93	0.00168	4.9
390.6	4526	0.54	0.0807	−0.5	388.7	3640	1.89	0.0703	−1.8
418.4	4373	7.15	1.14	0.4	383.8	4280	1.34	0.0421	−1.0
416.5	4213	5.82	0.961	0.4	378.9	4161	0.76	0.0245	−2.1
415.0	4283	5.21	0.846	1.2	374.0	4320	0.47	0.0145	−0.7
424.1	4333	11.01	1.79	−4.6	369.1	6880	0.45	0.00863	3.9
420.7	3227	6.45	1.40	1.0	364.2	12,920	0.47	0.00472	1.1
417.8	3762	5.78	1.07	−0.1	359.3	35,846	0.66	0.00238	−8.2
5-amino-1,10-phenanthroline									
*∮*/mm = 1					*∮*/mm = 3				
441.9	1731	0.92	0.407	−0.5	402.4	6685	0.44	0.00979	2.7
438.7	3251	1.32	0.312	0.9	399.3	6935	0.33	0.00706	4.6
436.3	4643	1.54	0.254	2.3	396.6	8429	0.30	0.00519	3.4
433.6	1920	0.50	0.200	3.2	393.7	10,468	0.27	0.00375	3.6
427.6	4411	0.66	0.113	2.1	390.5	18,821	0.32	0.00245	−2.9
421.0	5136	0.40	0.0584	0.2	387.1	28,236	0.33	0.00168	−1.9
418.3	5221	0.31	0.0449	0.7	384.4	45,736	0.42	0.00132	5.3
415.4	7575	0.32	0.0319	−5.4	381.4	158,054	1.01	0.000919	5.6
412.1	4566	0.14	0.0229	−4.5	403.6	2880	0.21	0.0106	−2.0
409.0	6392	0.17	0.0193	9.5	400.9	3242	0.17	0.00775	−3.5
406.1	23,930	0.40	0.0123	−5.5	397.6	7321	0.28	0.00567	1.3
403.3	24,877	0.31	0.00906	−5.3	394.6	13,323	0.36	0.00393	−1.8
440.9	1680	0.81	0.373	−0.7	391.8	18,483	0.35	0.00277	−5.1
438.1	1500	0.56	0.286	−1.4	388.7	36,843	0.49	0.00195	−5.5
434.7	2721	0.75	0.210	−2.3	385.8	54,628	0.55	0.00146	−0.5
431.9	1540	0.33	0.161	−2.2	383.9	69,853	0.54	0.00112	−4.3
429.0	1400	0.23	0.123	−2.1					
425.9	1680	0.20	0.0920	−2.4					
423.2	1521	0.14	0.0706	−2.5					
416.9	3481	0.19	0.0416	6.8					
413.9	6120	0.25	0.0304	5.2					
410.4	8611	0.24	0.0203	1.2					
407.7	15,540	0.32	0.0154	0.8					
405.1	24,605	0.39	0.0116	0.1					

^a^ Estimated uncertainties (Standard uncertainties. Type B): u(*T*) = 0.2 K, u(*m*) = 0.01 mg and u(*p*) = 0.05*p*. Note that pressures are deliberately given with one more digit than is significant. ^b^ Δ*p*/*p* = (*p* − *p_calc_*)/*p*, where *p_calc_* is calculated from the fitting lines reported in Table 5.

**Table 4 entropy-24-00192-t004:** Solution and solvation enthalpies in benzene of substituted 1,10-phenanthrolines and their sublimation enthalpies at 298.15 K.

Compound	Δ_soln_*H*_m_^*A*_i_^^/S a^	Δ_solv_*H*_m_^*A*_i_^^/S b^	Δ_cr_^g^*H*_m_^0 c^
	kJ·mol^−1^	kJ·mol^−1^	kJ·mol^−1^
5-Cl-1,10-phenanthroline (cr)	19.39 ± 0.11	91.5 ± 1.0	110.9 ± 1.0
5-CH_3_-1,10-phenanthroline (cr)	18.34 ± 0.20	88.9 ± 1.0	107.2 ± 1.0

^a^ From Appendix A. ^b^ Calculated according to Equation (2). ^c^ Calculated according to Equation (1). Uncertainties of vaporization/sublimation enthalpy are twice standard deviation [15].

**Table 5 entropy-24-00192-t005:** Regression parameters of the temperature dependence of vapor pressure for the crystalline 5-substituted-1,10-phenanthroline derivatives.

Compound	Δ*T*/K	ln(*p*/Pa) = *A* − *B*/*T*		
		*A* ^a^	*B*/K ^a^	OD ^b^, *∮*/mm
5-Cl-1,10-phenanthroline	332.2–367.1	35.150 ± 0.245	13472 ± 85	1
	308.0–339.6	34.986 ± 0.401	13392 ± 129	3
5-CH_3_-1,10-phenanthroline	331.0–372.3	33.691 ± 0.227	12921 ± 80	1
	311.4–338.9	33.932 ± 0.296	12971 ± 96	3
5-CH_3_O-1,10-phenanthroline	348.5–372.3	40.905 ± 0.480	15864 ± 173	1
	325.0–352.6	41.114 ± 0.587	15904 ± 199	3
5-CN-1,10-phenanthroline	377.1–413.7	38.031 ± 0.290	15833 ± 115	1
	345.3–378.3	37.624 ± 0.313	15624 ± 113	3
5-NO_2_-1,10-phenanthroline	389.0–424.1	37.259 ± 0.112	15536 ± 45	1
	352.6–392.2	38.004 ± 0.214	15796 ± 80	3
5- NH_2_-1,10-phenanthroline	403.3–441.9	38.349 ± 0.274	17341 ± 116	1
	381.4–403.6	38.736 ± 0.574	17462 ± 225	3

^a^ The associated uncertainties are standard deviations of the fitted parameters. ^b^ OD = Orifice diameter.

**Table 6 entropy-24-00192-t006:** Sublimation enthalpies at the mean experimental temperature, and the corresponding values adjusted to 298.15 K. Uncertainties are calculated according to the formula: u = 1/*n*·√Σu_i_^2^, where u_i_ represent the uncertainties associated to all terms used to calculate these values and *n* the number of replicates (*n* = 2 in this case). The standard molar sublimation enthalpy recommended values are displayed in bold.

Compound	Method, OD	⟨*T*⟩/K	Δ_cr_^g^*H*_m_^0^(⟨*T*⟩)	*C*_p,m_^0^ (cr) ^b^	−Δ_cr_^g^*C*_p,m_^0^	Δ_cr_^g^*H* _m_^0^ (298.15 K) ^c^
			kJ·mol^−1^	J·K^−1^·mol^−1^	J·K^−1^·mol^−1^	kJ·mol^−1^
5-Cl-1,10-phenanthroline	KEML, 1 mm	349.6	112.0 ± 0.7	221.5	34.0	113.8 ± 0.9
	KEML, 3 mm	323.1	111.4 ± 1.1	221.5	34.0	112.2 ± 1.1
	average					*113.0 ± 0.7*
	SC ^a^					110.9 ± 1.0
	**recommended**					**112.0 ± 0.8**
5-CH_3_-1,10-phenanthroline	KEML, 1 mm	353.2	107.4 ± 0.7	229.4	35.2	109.4 ± 0.9
	KEML, 3 mm	324.8	107.8 ± 0.8	229.4	35.2	108.8 ± 0.9
	average					*109.1 ± 0.6*
	SC ^a^					107.2 ± 1.0
	**recommended**					**108.2 ± 0.8**
5-CH_3_O-1,10-phenanthroline	KEML, 1 mm	360.4	131.9 ± 1.4	279.2	42.6	134.6 ± 1.7
	KEML, 3 mm	338.6	132.2 ± 1.7	279.2	42.6	134.0 ± 1.8
	average					*134.3 ± 1.2*
	**recommended**					**134.3 ± 1.2**
5-CN-1,10-phenanthroline	KEML, 1 mm	395.0	131.6 ± 1.0	235.1	36.0	135.1 ± 1.5
	KEML, 3 mm	361.7	129.9 ± 0.9	235.1	36.0	132.2 ± 1.2
	average					*133.7 ± 1.0*
	**recommended**					**133.7 ± 1.0**
5-NO_2_-1,10-phenanthroline	KEML, 1 mm	405.3	129.2 ± 0.4	248.9	38.1	133.3 ± 1.4
	KEML, 3 mm	373.0	131.3 ± 0.7	248.9	38.1	134.2 ± 1.2
	average					*133.7 ± 0.9*
	**recommended**					**133.7 ± 0.9**
5- NH_2_-1,10-phenanthroline	KEML, 1 mm	422.5	144.2 ± 1.0	214.4	32.9	148.3 ± 1.7
	KEML, 3 mm	392.6	145.2 ± 1.9	214.4	32.9	148.3 ± 2.1
	average					*148.3 ± 1.4*
	**recommended**					**148.3 ± 1.4**

^a^ SC= Solution Calorimetry. ^b^ Calculated according to the group additivity procedure proposed by Chickos et al. [29,30,31,32] with group contribution values reported in [31]. ^c^ Δ_cr_^g^*C*_p,m_^0^/J·K^−1^·mol^−1^ = −[0.75 + 0.15 *C*_p,m_^0^(cr)] [29,30,31,32].

**Table 7 entropy-24-00192-t007:** Vapor pressures at the average temperatures, *p*(⟨*T*⟩), sublimation entropies at the mean experimental temperature and pressure, Δ_cr_^g^*S*_m_(⟨*T*⟩, *p*(⟨*T*⟩), and the corresponding standard sublimation entropies, enthalpies and Gibbs energies adjusted to 298.15 K.

Compound	*p*(⟨*T*⟩)/Pa	Δ_cr_^g^*S*_m_(⟨*T*⟩, *p*(⟨*T*⟩)	Δ_cr_^g^*S*_m_^0^(298.15 K, *p*°)	Δ_cr_^g^*H*_m_^0^(298.15 K)	Δ_cr_^g^*G*_m_^0^(298.15 K)
		J·K^−1^·mol^−1^	J·K^−1^·mol^−1^	kJ·mol^−1^	kJ·mol^−1^
5-Cl-1,10-phenanthroline	0.0341 ^a^	320.3 ± 2.0	201.9 ± 4.3	113.8 ± 0.9	53.5 ± 2.2
	0.00156 ^b^	344.6 ± 3.3	197.9 ± 5.9	112.2 ± 1.1	53.2 ± 2.9
		average ^c^	*199.9 ± 3.7*	*113.0 ± 0.7*	*53.4 ± 1.8*
5-CH_3_-1,10-phenanthroline	0.0554 ^a^	304.2 ± 1.9	190.4 ± 4.1	109.4 ± 0.9	52.6 ± 2.2
	0.00247 ^b^	332.0 ± 2.5	189.4 ± 4.5	108.8 ± 0.9	52.3 ± 2.2
		average ^c^	*189.9 ± 3.0*	*109.1 ± 0.6*	*52.5 ± 1.6*
5-CH_3_O-1,10-phenanthroline	0.0446 ^a^	366.0± 4.0	252.5 ± 7.5	134.6 ± 1.7	59.3 ± 3.9
	0.00287 ^b^	390.5 ± 4.9	251.5 ± 8.7	134.0 ± 1.8	59.0 ± 4.3
		average ^c^	*252.0 ± 5.7*	*134.3 ± 1.2*	*59.1 ± 2.9*
5-CN-1,10-phenanthroline	0.1286 ^a^	333.2 ± 2.4	230.6 ± 5.9	135.1 ± 1.5	66.4 ±3.3
	0.00378 ^b^	359.2 ±2.6	224.1 ± 5.4	132.2 ± 1.2	65.4 ± 2.8
		average ^c^	*227.3 ± 4.0*	*133.7 ± 1.0*	*65.9 ± 2.2*
5-NO_2_-1,10-phenanthroline	0.3419 ^a^	318.6 ± 0.9	225.5 ± 4.8	133.3 ± 1.4	65.9 ± 2.9
	0.0130 ^b^	352.2 ± 1.8	228.8 ± 4.6	134.2 ± 1.2	66.0 ± 2.5
		average ^c^	*227.3 ± 3.3*	*133.7 ± 0.8*	*66.0 ± 1.9*
5- NH_2_-1,10-phenanthroline	0.0679 ^a^	341.2 ± 2.3	234.6 ± 6.5	148.3 ± 1.7	78.3 ± 3.6
	0.00322 ^b^	369.8 ± 4.8	235.4 ± 9.2	148.3 ± 2.1	78.1 ± 4.9
		average ^c^	*235.0 ± 5.6*	*148.3 ± 1.4*	*78.2 ± 3.0*

^a^ OD = 1 mm (series B); ^b^ OD = 3 mm (series A); ^c^ Uncertainty of the averages of the standard sublimation entropies, enthalpies and Gibbs energies (u_av_) are calculated by combining the single uncertainties of the two series (u_A_ and u_B_, respectively) according to the following expression: u_av_ = ½·[(u_A_)^2^ + (u_B_)^2^] ^0.5^.

## Data Availability

Data is contained within the article or Appendix A.

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
