# Peer review of "Sublimation Study of Six 5-Substituted-1,10-Phenanthrolines by Knudsen Effusion Mass Loss and Solution Calorimetry"

_entropy, 2022, doi:10.3390/e24020192_

Round 1
Reviewer 1 Report
The paper contains new experimental and calculated data, corresponds to the profile of the journal and can be published after minor revision in accordance with the recommendations below:
- Please, give the affiliations 1, 2, 3 and 5 in English, in the paper and in supplementary material.
- Abstract, line 27-29: “For this purpose, solution enthalpies at infinite dilution of studied phenantrolines in benzene were measured at 298.15 K” – please indicate two phenantrolines for which the enthalpies of dissolution at infinite dilution are determined in this article. The same recommendation is for the lines 153 and 192.
- The Table S2 should be transferred from supplementary material to the main text. Please check the indications a,b and c from the legend of the Table S2. The average values of ∆solnHmAj/S in tables S2 and S1 should be the same, as well as a values and symbols. Otherwise, their slight differences would have to be explained.
- References to recently published articles are welcome.
Author Response
General remarks
All the authors wish to thanks both Reviewers for their valuable work and for providing useful comments and requests of changes. In this document we attempted to answer to each query (point by point, in red). In our opinion the quality of the revised version is significantly improved hoping the current version of the manuscript is now suitable to be published in Entropy
Reviewer 1
- Please, give the affiliations 1, 2, 3 and 5 in English, in the paper and in supplementary material.
We made this correction following the Reviewer’s suggestion. - Abstract, line 27-29: “For this purpose, solution enthalpies at infinite dilution of studied phenantrolines in benzene were measured at 298.15 K” – please indicate two phenantrolines for which the enthalpies of dissolution at infinite dilution are determined in this article. The same recommendation is for the lines 153 and 192.
We thank the Reviewer for pointing out this issue. We revised accordingly. - The Table S2 should be transferred from supplementary material to the main text. Please check the indications a,b and c from the legend of the Table S2. SVC The average values of ∆solnHmAj/S in tables S2 and S1 should be the same, as well as a values and symbols. Otherwise, their slight differences would have to be explained.
We thank the Reviewer for pointing out this issue. We revised accordingly. Table S2 of the original manuscript is now Table 4 of the current version, and the other tables were renumbered. - References to recently published articles are welcome.
We are aware that the references are mostly related to papers published by more than five years. So, we made all the efforts by searching relevant publications in the last 2-5 years or other references that we missed in the preparation of the original manuscript. Actually, we found (and added in the revised manuscript) nine relevant papers that deserve to be cited, seven of which published in 2021. We added their citation in the revised version.
Reviewer 2 Report
Sublimation study of six 5-substituted-1,10-phenanthrolines by Knudsen Effusion Mass Loss and solution calorimetry
Bruno Brunetti, Andrea Ciccioli, Andrea Lapi, Aleksey V. Buzyurov, Ruslan N. Nagrimanov, Mikhail A. Varfolomeev,* and Stefano Vecchio Ciprioti
The authors report the sublimation enthalpy and entropy of several derivatives of 1,10-phenathroline and compare their results to those evaluated by solution calorimetry. The agreement is quite good. The work appears to have been carefully performed. I have made a number of comments that are included on the manuscript also being returned. Some of the comments are suggested; most are editorial comments. I also include a comment below that the authors should consider.
The nitrile group is linear. Ignoring partial double bond formation, exactly much does rotation of two individual atoms along a linear molecular axis contribute to the entropy compared to revolutions of mass relative to a central point such as occurs in the rotation of a NH2 group ? Assuming the phenanthroline ring is electron donating in the ground state, it would have opposing effects on rotation of the nitrile and amino group. Similarly if it were electron withdrawing. Since the two groups are different, I am not sure a comparison of sublimation entropy is meaningful here especially since the amino group is most likely hydrogen bonded in the solid state. If “rotation” of the nitrile contributes to the total entropy of sublimation, what would prevent it from doing so in the solid state as well?

Author Response
General remarks
All the authors wish to thanks both Reviewers for their valuable work and for providing useful comments and requests of changes. In this document we attempted to answer to each query (point by point, in red). In our opinion the quality of the revised version is significantly improved hoping the current version of the manuscript is now suitable to be published in Entropy
Reviewer 2
The nitrile group is linear. Ignoring partial double bond formation, exactly much does rotation of two individual atoms along a linear molecular axis contribute to the entropy compared to revolutions of mass relative to a central point such as occurs in the rotation of a NH2 group ?
Assuming the phenanthroline ring is electron donating in the ground state, it would have opposing effects on rotation of the nitrile and amino group. Similarly if it were electron withdrawing. Since the two groups are different, I am not sure a comparison of sublimation entropy is meaningful here especially since the amino group is most likely hydrogen bonded in the solid state. If “rotation” of the nitrile contributes to the total entropy of sublimation, what would prevent it from doing so in the solid state as well?
We agree with the remark of the referee. In the revised version, the analysis of the possible entropic effect related to the rotation of the substituent was reshaped and the explicit mention of the case of nitrile, to which the argument does not apply, was added. While an accurate evaluation of the entropic contribution of this rotation would require the theoretical evaluation of rotational barriers (which is beyond the scope of our work), a brief tentative discussion of the rather large differences in the sublimation entropies in the light of a supposedly enhanced rotation in the gas phase was retained because it seems appropriate.
As far as the comment in the pdf annotation file (lines 292-293) is concerned, this statement was reshaped by detailing the pressure ratio and by specifying that the reported pressure ratio refers to the temperature range explored in the experiments. Please, note that due to the very similar Clausius-Clapeyron slopes for the three compounds (see Figure 3), the pressure ratio changes only marginally with the temperature.
As far as the comment in the pdf annotation file (line 300) is concerned, we refer here to either chloro- and methyl- derivatives, which are the most volatile of the entire series with very similar partial pressures in the explored temperature range (see Figure 3). We changed the text accordingly.